# Effects of Physical Interventions on Subjective Tinnitus, a Systematic Review and Meta-Analysis

**DOI:** 10.3390/brainsci13020226

**Published:** 2023-01-29

**Authors:** Eric J. Bousema, Elouise A. Koops, Pim van Dijk, Pieter U. Dijkstra

**Affiliations:** 1Department of Otorhinolaryngology, Head & Neck Surgery, University of Groningen, University Medical Centre Groningen, 9712 CP Groningen, The Netherlands; 2Fysiotherapie Sittard Oost, 6137 RX Sittard, The Netherlands; 3Graduate School of Medical Sciences, Research School of Behavioural and Cognitive Neurosciences, University of Groningen, 9712 CP Groningen, The Netherlands; 4Gordon Center for Medical Imaging, Department of Radiology, Massachusetts General Hospital, Harvard Medical School, Boston, MA 02115, USA; 5Department of Oral and Maxillofacial Surgery, University of Groningen, University Medical Centre Groningen, 9712 CP Groningen, The Netherlands; 6Department of Rehabilitation Medicine, University of Groningen, University Medical Centre Groningen, 9712 CP Groningen, The Netherlands

**Keywords:** tinnitus, physical interventions, TENS, laser therapy, acupuncture, physical therapy

## Abstract

Increasingly, patients suffering from subjective tinnitus seek help from physical therapists. Numerous randomised controlled trials (RCTs) have investigated the effect of physical interventions commonly used in physical therapy practice on subjective tinnitus. This systematic review and meta-analysis aimed to analyse the effects of physical interventions on tinnitus loudness, tinnitus annoyance, and scores on the Tinnitus Handicap Index (THI). Four databases were searched from inception up to March 2022. A total of 39 RCTs were included in the systematic review, and 23 studies were appropriate for meta-analyses. Risk of bias assessments were also performed. Interventions analysed in at least five studies were summarised, including transcutaneous electrical nerve stimulation (TENS), laser therapy, and acupuncture. Random-effects meta-analysis models were used, and effect sizes were expressed as Hedge’s standardised mean differences (SMD) with 95%CI’s. The quality of three-quarters of the studies was limited due to insufficient allocation concealment, lack of adequate blinding, and small sample sizes. Large, pooled effects sizes were found for acupuncture (SMD: 1.34; 95%CI: 0.79, 1.88) and TENS (SMD: 1.17; 95%CI: 0.48, 1.87) on THI as well as for acupuncture on tinnitus loudness (VAS Loudness (SMD: 0.84; 95%CI: 0.33, 1.36) and tinnitus annoyance (SMD: 1.18; 95%CI: 0.00, 2.35). There is some evidence that physical interventions (TENS and acupuncture, but not laser therapy) may be effective for tinnitus. However, the lack of high-quality studies and the risk of bias in many studies prohibits stronger conclusions.

## 1. Introduction

Subjective tinnitus is a common symptom that is characterised by the perception of a phantom sound not caused by an external acoustic stimulus [1]. The prevalence of subjective tinnitus in the general population ranges from 12 to 30% [2]. Although the prevalence increases with both hearing loss and increasing age [3,4,5], it is a problem for all age groups. The impact of tinnitus can be very mild, but in some cases, it can severely impact the quality of life [3,6,7].

Although currently available treatments may relieve tinnitus completely in some individual cases [8,9], a treatment that cures tinnitus in a large proportion of tinnitus suffers, largely due to its multifactorial aetiology, does not exist. A large number of randomised controlled trials (RCTs) have been performed to investigate treatment options, ranging from hearing aids to physical, psychological, and drug-related interventions. The majority of these treatments aim to ameliorate the tinnitus percept or aim to reduce the negative impact of subjective tinnitus on the quality of life [10].

Increasingly, patients suffering from subjective tinnitus seek help from physical therapists who apply physical interventions such as transcutaneous electrical nerve stimulation (TENS), laser therapy, temporomandibular disorder (TMD) treatment, acupuncture, biofeedback, and kinesio taping (see Box 1). The rationale behind these physical interventions for tinnitus is not always well-substantiated, such as acupuncture, however, in a number of interventions, such as transcutaneous electrical nerve stimulation (TENS), laser therapy, and temporomandibular disorder (TMD) treatment, the somatosensory stimuli may affect the auditory system through a connection between the cochlear nucleus (CN) and the trigeminal and dorsal column systems of the somatosensory system [11]. A consequence of this presumed connectivity is that two-thirds of the patients with subjective tinnitus can modulate the loudness and pitch of their tinnitus, which is presumably mediated via these interconnected systems, by contracting their neck, head, or jaw muscles [12,13,14,15,16,17,18,19]. These connections may also explain why patients with subjective tinnitus have an average of 2.6- and 6.7-times greater risk of reporting cervical spine disorders (CSD) or temporomandibular disorders (TMD), respectively [20,21,22].

The RCTs that analysed the effects of these physical interventions focused mainly on subjective tinnitus loudness and annoyance as outcome measures. Some studies reported significant effects [23,24], whereas others did not [25,26]. In these trials, a great diversity of interventions were investigated, which likely contributed to the large variation in reported results. In 2016, a systematic review regarding the effects of physical therapy on subjective tinnitus was published without meta-analysis [27]. Since that review, several new studies have been published. Recently, a systematic review regarding laser therapy and subjective tinnitus was published, but it too was without meta-analysis [28]. In 2021, a systematic review and meta-analysis regarding acupuncture and subjective tinnitus was published but they only searched up to September 2018 [29]. A current overview and an estimation of the pooled effects of physical interventions will help clinicians to select appropriate treatments for subjective tinnitus and may inform the research community regarding knowledge gaps.

Therefore, with this systematic review and meta-analysis, we aimed to investigate the effects and the mutual degree of effectiveness of physical interventions in patients with subjective tinnitus on measures of tinnitus perception.

Box 1Description of physical interventions commonly used in physical therapy practice on subjective tinnitus.
**Physical interventions**
**TENS** for tinnitus consists of placing an electrode around the mastoid or at the arm. Stimulation of the median nerve or the temporomandibular joint region aims to modulate tinnitus [30].**Laser therapy** for tinnitus is often focused via the external ear canal on the inner ear. The rationale behind this therapy is that increasing the blood flow and accelerating the metabolism may improve cochlear function [31,32].**Acupuncture therapy (AP)** for tinnitus is performed by placing needles around the ear and at more peripheral locations, sometimes in conjunction with electrical stimulation (eAP). Neurophysiological modulation of the autonomic nervous systems, the olivocochlear nucleus, the non-classical ascending auditory pathway, and the somatosensory system are suggested as possible mechanisms underlying the treatment of tinnitus by acupuncture [33,34].**TMD Treatment (Orofacial therapy)** consists of treating muscles of the face, jaw, and neck with massages and stretching, and provides instructions for a relaxed resting position of the mandible. It may affect the auditory system through a connection of the cochlear nucleus (CN) with the trigeminal and dorsal column systems of the somatosensory system [11].**Kinesio tape** is an elastic cotton adhesive strip applied to the neck and shoulder region and aims to relieve tension in the neck area, reduce trigger points, provide mobility to the tissue, and improve blood and lymphatic flow [35,36].**Biofeedback** is a relaxation technique via electromyography feedback to decrease muscle tension related to stress. It may help to reduce tinnitus annoyance [37].

## 2. Materials and Methods

### 2.1. Systematic Search Strategy and Study Selection

The database search in PubMed, Cochrane library, Pedro, and EMBASE was conducted from inception to March 2022. The main topics of the search strategy were physical therapy modalities, physiotherapy, musculoskeletal, spine, biofeedback, manual, trigger point, and exercise (Figure A1). Studies were included if they concerned an RCT and if a physical intervention commonly used in physical therapy practice was analysed for its effects on perceived tinnitus. Studies were excluded if they concerned reviews, letters to the editor, case reports, and study protocols. Studies not written in English or Dutch were excluded for language reasons. Titles and abstracts were assessed for their relevance independently by three observers (E.K., E.B., and P.U.D.). In the next round, full-text manuscripts were assessed by the same observers. References of the studies included were checked for relevant studies that had been missed in the database searches. All selected studies were assessed for risk of bias by the same observers, making use of the Cochrane Collaboration tool (E.K., E.B., and P.U.D.) [38]. This study is registered at Prospero (ID: CRD42022303775).

### 2.2. Meta-Analysis

A wide range of outcome measures was applied by the various studies to measure perceived tinnitus. The meta-analysis was performed with outcome measures reported in at least five studies. Measurements made using a visual analogue scale (VAS) were pooled into two categories: VAS loudness and VAS annoyance. Data reported as VAS intensity and VAS loudness were grouped under VAS loudness. Data reported as VAS distress, VAS uncomfortableness, VAS discomfort, and VAS annoyance were grouped under VAS annoyance. Based on the research question in the corresponding study, VAS severity was assigned to either VAS loudness or VAS annoyance. In addition to the VAS scores, the “Tinnitus Handicap Index” (THI) [39] was another frequently used outcome measure. For these three outcome measures, separate meta-analyses were performed. Studies were excluded from the meta-analysis if their outcome measures did not fit one of the three grouped outcome categories. For studies with insufficient data for meta-analysis, we contacted the corresponding author to request additional information by email. We did not contact authors from studies published more than 25 years ago.

The types of interventions that were investigated in these selected studies included laser therapy, TENS, acupuncture, TMD treatment, kinesio tape, and biofeedback (see box). Laser therapy, TENS, and acupuncture were investigated in a sufficient number of studies and therefore included in the meta-analysis. Interventions that were investigated in a limited number of studies were consequently not included in the meta-analysis and will be presented under the collective category “Other interventions”.

The meta-analysis results are presented for each outcome variable differentiated per intervention. In some studies, the intervention was compared to another active intervention instead of a sham treatment or waiting list group. These results will be presented separately.

### 2.3. Data Synthesis and Analysis

Data were entered in the computer program Comprehensive Meta-Analysis V3 (Biostat, Englewood, NJ, USA). In this review, many small sample studies were included. The effect size Cohen’s d overestimates the effects of small studies [40]. We, therefore, used Hedge’s standardised mean differences (SMD) and 95% confidence intervals (CI) as effect size since it corrects this overestimation. Meta-analyses using a random-effects model were performed when two or more studies investigated the same interventions due to the clinical and methodological heterogeneity of the included studies. Meta-analyses have a greater power to detect differences in effects due to the pooling of data. Meta-regression was performed to analyse the effects of duration upon follow-up after ending treatment, the type of control intervention, and the year of publication on the outcome measures. None of the studies reported a correlation of the mean between pre- and post-data, which is necessary to differentiate covariance effect size from real effect size. Based on data received from one of the contacted authors, we were able to calculate the correlation between pre- and post-data regarding VAS annoyance (0.659) [25]. This value was imputed for all studies since no other data was available. We consider effect sizes of 0.2–0.5 as small, 0.5–0.8 as medium, and 0.8–1.0 as large [41]. For each domain of the risk of bias tool, a meta-regression was performed to explore the effect of the different domains of bias on the outcome measures (Table A1).

## 3. Results

### 3.1. Search Strategy and Study Selection

A total of 1127 records were identified, of which 315 were in PubMed, 414 in Embase, 359 in Cochrane, and 39 in Pedro. After the assessment of their eligibility, 39 studies were included in the systematic review, and 23 studies were summarised in the meta-analyses (Figure 1: flow diagram).

### 3.2. Descriptive Study Characteristics

A total of 3120 patients with tinnitus participated in the included studies. On average, the mean number of patients in each study was 53. The included studies reported on various intervention types and intensities, and measuring instruments (Table 1, Table 2, Table 3 and Table 4, overview of studies).

### 3.3. Risk of Bias

The inter-observer reliability of the risk of bias assessment was substantial (Cohen’s Kappa 0.76 [73]). Five studies were rated to have a high risk of bias due to a lack of adequate randomisation, and eighteen studies did not report the randomisation clearly (Table 5). Furthermore, three studies did not conceal allocation and twelve studies did not describe allocation concealment. In seventeen studies, the participants, study personnel, or assessor were not blinded. Two studies did not fully report outcome data. In eighteen studies, no information regarding dropouts was reported or recorded as unclear. In terms of selective outcome reporting, the risk of bias was assumed in five studies. In three studies, only the significance levels or graphics were reported [58,70]. For one of these studies, additional information was received after contacting the author [25]. For twenty-two studies, the risk of bias was rated as unclear as no study protocol was available. Regarding other sources of bias, two studies were rated unclear due to uncertainty regarding generalisability because only male subjects were included [51] and unclear suitability of the sham condition, as false meridian points around the ear could have physiological effects [61]. In three studies, the risk of bias was present because of excluding patients [67], unequal groups [50] and inappropriate statistical methods [64]. The meta-regression to explore the effect of these different domains of bias on the outcome measures showed that for VAS loudness, blinding, and selective outcome reporting, the risk of bias significantly differed from 0 (Table A1(c)).

### 3.4. Intervention, Intensity, and Location

The shortest intervention was a 15-s single session of acupuncture [58]. The intervention with the longest duration consisted of 20-min daily laser treatments for the duration of 10 weeks [49]. The vast majority of interventions (n = 24) were applied in the vicinity of the ear, external auditory meatus, and mastoid process. In some cases (n = 6) of acupuncture, the intervention was applied to more distal regions such as the arms or legs. In most studies, the reported control interventions were either a sham treatment (n = 29) or a waiting list (n = 4) [24,26,37,63]. Seven studies compared the intervention of interest with another active intervention [13,53,64,66,68,69,72].

### 3.5. Outcome Measures

Most instruments could be clustered into one of the following three outcome measures (Figure 2, Figure 3 and Figure 4): VAS loudness (21 studies), VAS annoyance (17 studies), and THI (22 studies). In addition to these three outcome measures, VAS awareness/attention (5 studies), Tinnitus Questionnaire (3 studies), and tinnitus matching (4 studies) were frequently used instruments (Table 1, Table 2, Table 3 and Table 4).

### 3.6. Physical Interventions

#### 3.6.1. TENS

In six studies, Transcutaneous Electrical Nerve Stimulation (TENS) was analysed (Table 1). In two studies, the TENS treatment effect was larger than that of the sham treatment (*p* < 0.05) [30,45]. In two other studies, the treatment effect between groups was not analysed [43,44]. One study [42] reported no effect of TENS. In the last study, TENS was compared to osteopathy and the authors reported a similarly helpful effect [13]. Four studies were included in the meta-analysis.

#### 3.6.2. Laser

Ten studies investigated the effects of laser therapy on tinnitus (Table 2). Four studies reported a larger effect of laser therapy than of the sham laser [48,49,51,54], while four other studies [25,47,50,52] did not find a difference in effects between the laser and sham treatment. In one study, a larger effect of sham treatment was found but not statistically substantiated [46]. In another study, two different laser treatments were compared, resulting in a significant difference in effectiveness [53]. Seven studies were included in the meta-analysis.

#### 3.6.3. Acupuncture

The effect of acupuncture on tinnitus was investigated in 15 studies (Table 3). Three studies analysed the effects of acupuncture and compared these effects to another form of acupuncture or TENS [59,64,66]. No significant differences were reported within these interventions. In ten other studies, a sham treatment was included, and in one study no treatment was given. Seven studies reported an effect of acupuncture intervention on tinnitus [23,34,43,51,68,69,70] whereas three studies did not find any effect [26,56,57] and two studies reported mixed significant outcomes [61,62]. Ten studies were included in the meta-analysis.

#### 3.6.4. Other Interventions

Eight studies investigated other physical interventions for tinnitus (Table 4). In three studies, the effect of TMD treatment on tinnitus was investigated [24,68,72]. One study found a significant effect of TMD treatment compared to placebo [24]. TMD treatment combined with manual therapy was more effective than TMD treatment alone [72]. TMD treatment was not more effective than biofeedback [68]. Two studies analysed the effects of biofeedback and found it to be more effective than Amitriptyline [69] and the waiting list [37]. Myofascial trigger point therapy [70] was more effective than a sham treatment. Ultrasound [67] had no effect compared to a sham treatment. Finally, one study showed significant improvement in the kinesio tape group but results were not analysed between the groups [71]. In two studies, two different interventions were compared [68,69]. A total of seven studies could be included in the meta-analysis.

### 3.7. Meta-Analysis Results

The results of the meta-analysis are presented in Figure 2, Figure 3 and Figure 4. To facilitate the comparison between interventions, each of these figures shows pooled effect sizes per intervention for a single outcome measure. Since the interventions included in the meta-analysis have different rationales, an overall pooled effect size for one outcome measure is not presented. Several studies did not contain appropriate data for a meta-analysis while for three of these studies, the data were obtained from the authors after submitting a data request [24,25,59]. In four other studies we were not successful [47,58,63,74]. Three studies were published more than 25 years ago, and therefore, the authors were not contacted [55,56,69]. Nine other studies were not included in the meta-analysis as their outcome measures could not be added to one of the derived categories of VAS loudness, VAS annoyance, or THI. In one study [65], the author reported such small standard deviations for the VAS loudness scores that we suspect these data were actually standard errors. As the author did not respond to our request for clarification, we performed one analysis with the standard deviations as reported by the authors (Figure A2) and one analysis with the standard deviations recalculated by us. In both calculations, the SMD showed a significant effect size and, consequently, we have presented the analysis reflecting our calculated standard deviation.

#### 3.7.1. VAS Annoyance

In nine sham-controlled studies, treatment effects on VAS annoyance were investigated. Effects of acupuncture (four studies [23,57,62,63]), laser (three studies [25,46,52]), and TENS (two studies [13,44]) were entered into the meta-analysis. A significant effect was observed for acupuncture (SMD: 1.18; 95%CI [0.00, 2.35]) (Figure 2A).

In eight non-sham controlled studies, one study compared biofeedback in combination with CBT to a waiting list group, showing a significant effect on tinnitus annoyance (SMD: 0.71; 95%CI [0.32, 1.09]) [37]. Two studies compared various types of acupuncture [64,66] and found no significant effects. One study reported that a combination of laser and rTMS was more effective in reducing tinnitus annoyance than each of these treatments alone [53]. Similarly, manual therapy in combination with TMD treatment was more effective than TMD treatment alone (SMD: 1.11; 95%CI [0.56, 1.67]) [72]. There was no significant difference between TENS treatment and osteopathy [13] or between biofeedback and amitriptyline [69] (Figure 2B).

#### 3.7.2. VAS Loudness

Thirteen sham-controlled studies analysed treatment effects on VAS Loudness. Acupuncture (three studies [57,60,65]) reduced VAS loudness (SMD: 0.84; 95%CI [0.33, 1.36]), while laser therapy (two studies) [46,50,51,52] and TENS (three studies [42,43,44]) did not significantly reduce VAS loudness. Effects of other treatments (three studies) were significant for two interventions: MTP treatment and kinesio taping (SMD: 0.92; 95%CI [0.37, 1.47] and SMD: 1.15; 95%CI [0.40, 1.91], respectively) [70,71], but not ultrasound [67] (Figure 3A).

In eight non-sham controlled studies, three studies compared a treatment group to a waiting list group which showed a significant effect for biofeedback plus CBT (SMD: 0.64; 95%CI [0.26, 1.02]) [37], but not for acupuncture or TMD treatment [24,26]. Five studies compared different types of acupuncture [64,66] or compared one type of treatment to another type of treatment [13,66,68,69] and reported no significant effects (Figure 3B).

#### 3.7.3. THI

In 15 sham-controlled studies, the effects of treatment on THI were analysed (Figure 4a). Acupuncture (five studies) was effective in reducing THI scores (SMD: 1.34; 95%CI [0.79, 1.88]) [23,34,61,62,63]. Similarly, TENS (three studies) was effective in reducing THI scores (SMD: 1.17; 95%CI [0.48, 1.87]) [30,44,45]. However, laser therapy (five studies) did not affect THI scores [46,49,50,51,52]. In two studies, MTP treatment [70] and kinesio taping [71] were effective in reducing THI scores (SMD: 0.92; 95%CI [0.37, 1.47] and SMD: 0.81; 95%CI [0.08, 1.54], respectively).

In seven non-sham controlled studies, one study compared biofeedback in combination with CBT to a waiting list group which showed a significant effect on THI score (SMD: 1.40; 95%CI [0.98, 1.81]) [37]. Two studies compared different types of acupuncture [64,66] and found no significant effect. A combination of laser and rTMS had a larger effect on THI score than each of these treatments alone [53], and manual therapy in combination with TMD treatment was more effective than TMD treatment alone (SMD: 1.65; 95%CI [1.05, 2.25]) [72]. Two studies showed no effects when comparing TENS treatment on either one or two ears with osteopathy [13,45] (Figure 4B).

## 4. Discussion

Over recent decades, a large number of RCTs have investigated the effect of physical interventions on subjective tinnitus. This systematic review and meta-analysis investigated the effects of several physical interventions in patients with subjective tinnitus on the outcome measures of THI, VAS annoyance, and VAS loudness.

Our meta-analyses showed that some physical interventions have an effect on VAS loudness, VAS annoyance, and THI. Acupuncture as a mono treatment had a significant effect on VAS loudness and VAS annoyance, and both acupuncture and TENS had a significant effect on THI scores. Several other treatments had a significant effect on the three outcome measures, but the number of studies was too few to draw firm conclusions.

The study results reported in this review and meta-analysis should be interpreted in light of the quality of the studies. The quality assessment showed that 11 studies had a low or unclear risk of bias, and we had some concerns regarding inadequate concealment and/or blinding in 28 studies. As most physical interventions can be felt or seen, these interventions are very hard to blind. However, laser therapy can be blinded more easily, though no significant effects of laser therapy on the outcome measures were found. Based on these results, there is insufficient evidence to support the use of laser therapy as an effective intervention for tinnitus. This conclusion is consistent with another recent systematic review of laser therapy for tinnitus [28].

Although the studies investigating TENS therapy showed a significant effect on THI, these studies were not blinded, or we were uncertain about adequate blinding. Interestingly, in this review, we did not find any studies investigating the effect of TENS on tinnitus reporting non-significant results. This lack of studies with no effect might indicate reporting bias. Therefore, the efficacy of TENS on tinnitus cannot be adequately assessed based on the available studies.

In acupuncture, blinding is also difficult as the placing of a needle is felt by the patient. Researchers tried to blind a control group by, for example, inserting the needle more superficially or away from effective acupuncture points; this might decrease the risk of bias. In addition, allocation concealment was often unclear, and the sample sizes were insufficient. However, statistical analysis of the risk of bias revealed no systematic effects of allocation concealment on the outcomes of the studies. Two systematic reviews and meta-analyses investigating the effects of acupuncture on tinnitus have already been published [29,75]. Similar to the current study, the latest systematic review reported that acupuncture might be beneficial based on the THI. In that systematic review, studies written in Chinese were also included. Due to the lack of high-quality studies, our results and those of previous reviews should be interpreted with some caution.

In the interventions that target areas around the ear, jaw, or neck, the effects on tinnitus may be explained by the normalisation of somatosensory input, which could be facilitated by the connection between the spinal trigeminal nucleus (Sp5) of the trigeminal nerve and the cochlear nucleus of the auditory system in the medulla oblongata [76]. For interventions that target areas outside of these aforementioned regions, such as acupuncture, the working mechanism is not clear [33,34]. It might be that acupuncture affects the autonomic nervous system in a manner similar to CBT and relaxation therapy [10].

Overall, based on the results of this review and meta-analysis, some physical interventions appear to be effective in reducing VAS annoyance and loudness scores and THI scores. This supports the concept that tinnitus not only can be modulated but can also be reduced in the long-term by normalisation of somatosensory input. In this review, we did not analyse the clinical relevance of the changes. Future research should analyse whether the effect sizes found are large enough to substantiate the application of the physical interventions studied.

It is crucial to realise that multiple systems and pathways may modulate tinnitus and may also interact with each other. Therefore, physical therapy for tinnitus may be more beneficial to the patient if multiple systems are targeted by the applied interventions. Very few studies that were included in this review investigated multimodal interventions, i.e., biofeedback combined with cognitive behaviour therapy or TMD treatment, a combination of education, exercises, and splint therapy [24,37]. These studies showed significance between group results but used other outcome measures than those included in this meta-analysis and were too few to draw a conclusion. However, these findings are supported by a systematic review of TMD treatment that also reported that many of these studies have methodological issues [27].

One of the strengths of the current systematic review and meta-analysis is that it summarises the effects of several physical interventions in a meta-analysis, allowing the comparison of the effect of those interventions. This comparison improves our understanding of the strengths and weaknesses of the treatments and thereby supports recommendations for future research investigating tinnitus treatments. Secondly, most studies included longitudinal data collection with multiple follow-ups. In this review, we considered the data of the latest follow-up, to prevent investigating short duration effects, as a long-lasting effect is clinically more meaningful.

A limitation of this study is that some potential existing evidence was not included in this review due to language reasons (e.g., excluding Chinese studies on the effect of acupuncture), which may have resulted in selection bias.

### Recommendations

The reported RCTs analysing the effects of physical interventions on subjective tinnitus are diverse in their outcome measures and methodological quality. Some domains of the risk of bias can be avoided easily, such as adequate reporting of dropouts, reporting actual outcomes with confidence intervals, and using outcome measures with known psychometric properties, thereby increasing the strength of the study. Studies using unvalidated or rare outcome measures generally cannot be included in a meta-analysis because there are too few studies with the same outcome measures. Even though these studies may be well-designed, they do not contribute to the level of evidence in this review. Therefore, we strongly recommend that in further research investigating the effect of physical interventions on tinnitus, the outcome measures, VAS loudness, VAS annoyance, and a validated tinnitus questionnaire (i.e., THI or Tinnitus Functional Index [4]), are included.

Not only is the mechanism of tinnitus complicated, but the factors modulating tinnitus are complex and often interdependent [77]. Some patients with tinnitus perceive high levels of psychological arousal [37] that influence the somatosensory system [78]. This arousal may lead to or enhance disorders in cranial or cervical regions and therefore can counteract the effects of physical interventions for tinnitus in these regions. In most of the included studies, psychological arousal as a cofounder was not taken into account. We recommend that in future research, the level of psychological arousal should be assessed for stratification.

## 5. Conclusions

Based on the results of this review, we conclude that there is some evidence that physical interventions such as TENS and acupuncture may be effective for tinnitus. The current lack of high-quality studies prohibits stronger conclusions, as caution is needed due to the presence of a risk of bias in many of the included studies. However, there is insufficient evidence to support laser therapy as a treatment for tinnitus.

## Figures and Tables

**Figure 1 brainsci-13-00226-f001:**
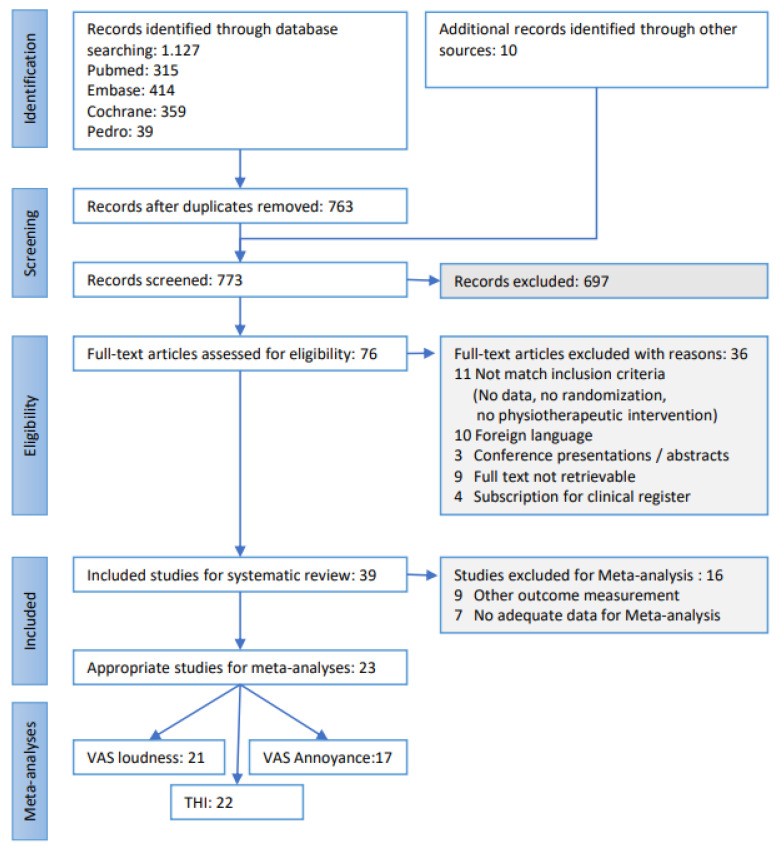
Flow chart of study selection.

**Figure 2 brainsci-13-00226-f002:**
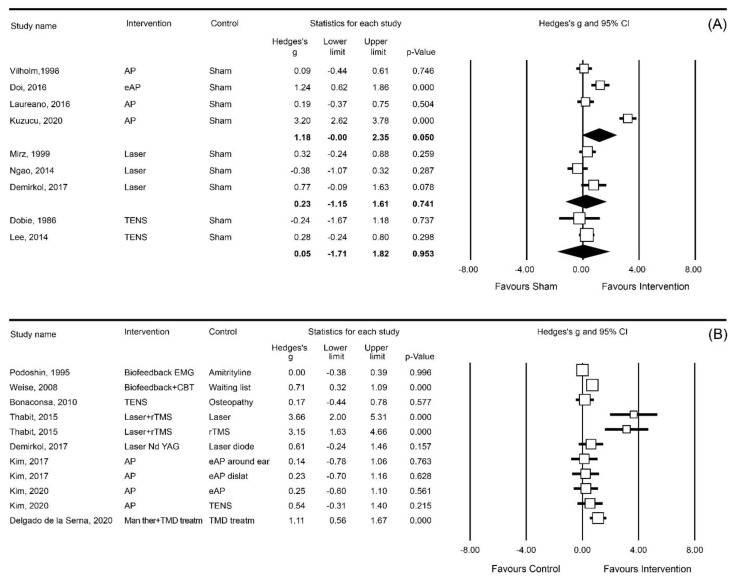
Meta-analyses of the effect of annoyance grouped by intervention: (**A**) sham-controlled; (**B**) non-sham controlled. Study name: studies reported double are three-arm RCTs. Intervention: rTMS = repetitive transcranial magnetic stimulation, AP = acupuncture, eAP = electro acupuncture, Treatm = treatment, Man ther = Manual therapy, and TMD = temporomandibular disorder. [13,23,37,42,44,46,52,53,54,57,62,63,64,66,69,72].

**Figure 3 brainsci-13-00226-f003:**
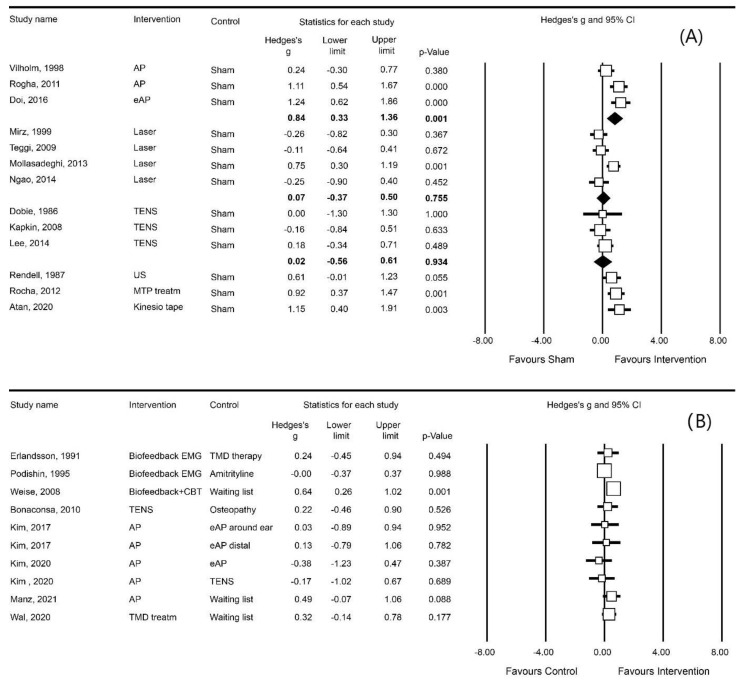
Meta-analyses of the effect on loudness grouped by intervention: (**A**) sham-controlled; (**B**) non-sham controlled. Study name: studies reported double are three-arm RCTs. Intervention: AP = acupuncture, eAP = electro acupuncture, Treatm = treatment, MTP = myofascial trigger points, TMD = temporomandibular disorder, CBT = Cognitive behaviour therapy, US = Ultrasound therapy, and Kinesio Tape = Kinesio tape therapy. [13,24,26,37,42,43,44,46,50,51,52,57,63,64,66,67,68,69,70,71].

**Figure 4 brainsci-13-00226-f004:**
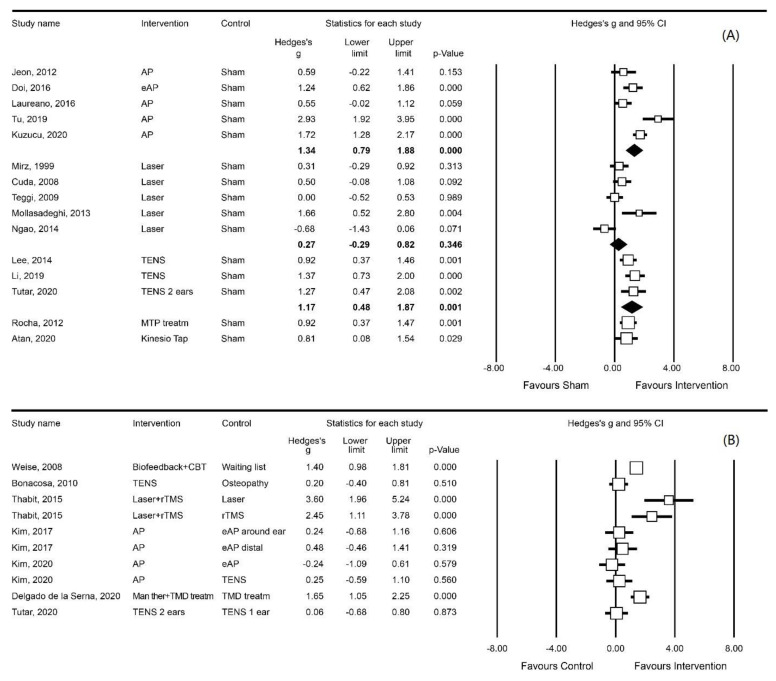
Meta-analyses of the effect on THI grouped by intervention: (**A**) sham-controlled; (**B**) non-sham controlled. Study name: studies reported double are three-arm RCTs. Intervention: rTMS = repetitive transcranial magnetic stimulation, AP = acupuncture, eAP = electro acupuncture, Treatm = treatment, MTP = myofascial trigger points, TMD = temporomandibular disorder, CBT = Cognitive behaviour therapy, Man ther = Manual therapy, and Kinesio Tap = Kinesio tape therapy. [23,30,34,44,45,45,46,49,50,51,52,61,62,63,64,66,70,71,72].

**Table 1 brainsci-13-00226-t001:** Overview of studies analysing the effect of TENS on tinnitus.

Author	Design	Intervention	N (♀)	Age (sd)	Control	N (♀)	Age (sd)	Location	Intensity	Outcomes of Interest				
											Intervention		Control	
											T0	T1	T0	T1
Dobie 1986 [42]	DB Cross	TENS	20 (5)	50 (12)	Sham	20 (5)	50 (12)	Mastoid	1 h to 24 h 7 days	L A		Cases 3/20 3/20	improved	2/20 3/20
Kapkin 2008 [43]	NB	TENS	31 (-)	45 (-)	Sham	11 (-)	45 (-)	Auricle Mastoid	3/week 1 month	L(dB) F(kHz)	7.2 (3.2) 5.8 (3.0)	7.3 (3.3) 5.8 (3.1)	7.2 (2.3) 6.6 (3.0)	6.6 (3.4) 5.9 (3.3)
Bonaconsa 2010 [13]	NB	TENS	20 (-)	49 (-)	Osteo-pathy	20 (-)	49 (-)	Cervical, upper back, Auricle	1/week 2 months	THI VAS L VAS A	51.4 (-)	36.3 (-) Mean −1.5 (-) −2.2 (-)	51.4 (-) change	42.9 (-) −0.5 (-) −1.4 (-)
Lee 2014 [44]	SB	TENS	45 (19)	47 (14)	Sham	20 (7)	46 (12)	Auricle	30 s/point 2/week, 4 weeks	THI VAS L VAS A	49.4 (9.9) 6.7 (1.7) 6.7 (1.5)	42.8 (8.7) 5.8 (1.9) 5.4 (2.2)	44.5 (6.5) 6.2 (1.9) 6.5 (1.7)	45.2 (7.9) 5.6 (1.6) 5.7 (2.2)
Li 2019 [13]	DB	TENS	23 (9)	49 (12)	Sham	23 (6)	48 (13)	C2	30 min, 3/week, 4 weeks	TH I **		Mean −11.6 {−15.3; −8.2}	change	−2.9 {−5.8; −1.5}
Tutar 2020 [45]	SB	TENS 1 ear	20 (-)	-	Sham	20 (-)	-	Auricle	30 min, 10 sessions	THI **	37.8 (20.7)	10.2 (8.9)	38.0 (21.0)	28.7 (15.0)
		TENS 2 ears	20 (-)	-					1 month	THI **	35.1 (21.1)	8.6 (4.9)		

Design: study design type: NB = not blinded, SB = single-blinded (participants), DB = double-blinded (personnel), Cross = Cross-over. Location: anatomic region where intervention is applied. Population: ♀ = number of woman. Outcomes of interest: THI = Tinnitus Handicap Inventory; VAS = Visual analogue scale 10 cm, L = Loudness, A = Annoyance, L (dB) = tinnitus loudness matching in decibels hearing level, F (kHz) = tinnitus frequency matching in kilohertz, () = Standard deviation, {} = 95% Confidence Interval, - = no data available, Between-group analysis: (**) *p* < 0.01.

**Table 2 brainsci-13-00226-t002:** Overview of studies analysing the effect of laser on tinnitus.

Author	Design	Intervention	N (♀)	Age (sd)	Controls	N (♀)	Age (sd)	Location	Intensity	Outcomes of Interest				
											Intervention		Control	
											T0	T1	T0	T1
Mirz 1999 [46]	DB	49 mW (729 nm)	25 (-)	-	Sham	25 (-)	-	external auditory meatus	10 min, 15 sessions	THI VAS L VAS A VAS At	39.8 (24.9) 6.6 (2.1) 5.8 (2.5) 6.8 (2.2)	38.8 (24.1) 6.6 (1.9) 6.6 (1.8) 7.0 (1.8)	45.7 (19.9) 6.7 (2.4) 6.3 (2.4) 6.9 (2.3)	38.7 (21.8) 6.2 (2.8) 6.3 (2.8) 6.0 (3.0)
Nakashima 2002 [47]	SB	60 mW (800 nm)	23 (12)	52 (12)	Sham	20 (15)	55 (14)	external auditory meatus	6 min 1/week, 4 weeks	NRS L NRS A		Nr’s of ears 7/33 6/33	improved	10/31 12/31
Gungor 2007 [48]	DB	5 mW (650 nm)	45 (-)	56 (-)	Sham	21 (-)	56 (-)	external auditory meatus	15 min, 7 sessions, 1 week	NRS L * NRS A **		Nr’s of ears 22/45 25/45	improved	4/21 4/17
Cuda 2008 [49]	NB	5 mW (650 nm) + counselling	26 (-)	50 (10)	Sham + counselling	20 (-)	64( 14)	external auditory meatus	20 min Daily, 3 months	THI *	53.6 (22.3)	36.6 (21.1)	43.1 (22.1)	35.8 (18.9)
Teggi 2009 [50]	DB	5 mW (650 nm)	27 (11)	52 (11)	Sham	27 (13)	53 (13)	external auditory meatus	20 min Daily, 3 months	THI VAS L MML	42.5 (24.2) 6.4 (2.2) 8.9 (5.7)	33.7 (26.1) 6.3 (2.4) 6.2 (3.4)	51.5 (36.6) 6.2 (2.0) 8.8 (5.4)	43 (24.2) 5.9 (2.3) 7.4 (4.3)
Dejakum 2013 [25]	DB	450 mW (830 nm)	19 (9)	57 (13)	Sham	22 (10)	50 (16)	external auditory meatus	30 min 12 sessions, 4 weeks	Goebel Q VAS L VAS A VAS At	Only graphs presented			
Mollasadeghi 2013 [51]	DB	5 mW (650 nm)	41 (0)	-	Sham	41 (0)	-	Proc. Mastoid	20 min every other day 20 sessions	L(dB) ** THI ** VAS L **	6.1 (1.1)	5.1 (1.2) Cases 14/41 12/41	6.1 (1.1) improved	6.0 (1.2) 1/41 1/41
Ngao 2014 [52]	DB	5 mW (650 nm) + betahistine	22 (13)	57 (-)	Sham + betahistine	21 (13)	59 (-)	external auditory meatus	20 min daily, 10 weeks	THI VAS L VAS A VAS Pitch		Cases 12/22 9/22 6/22 8/21	improved	17/21 11/22 9/21 6/21
Thabit 2015 [53]	DB	200 mW (808 nm) +	10 (3)	39 (14)	rTMS	10 (4)	41 (12)	around ear	260 s 10 sessions	THI * VAS A *	66.1 (7.1) 6.6 (0.8)	41.4 (7.5) 4.1 (0.6)	73.1 (4.3) 6.7 (0.5)	66.7 (4.6) 6.2 (0.4)
		rTMS			200 mW (808 nm)	10 (5)	36 (15)			THI * VAS A *			73.1(4.3) 6.7(0.5)	66.7(4.9) 6.2(0.5)
Demirkol 2017 [54]	SB	? (1064 nm)	15 (8)	37 (15)	Sham	15 (9)	38 (14)	external auditory meatus	9–20 s 5/week	VAS A *	5.0 {3.0–3.5}	Median 0 {0.0–2.0}	6.0 {4.0–8.0}	5.0 {4.0–7.0}
		? (810 nm)	16 (6)	40 (15)					10 weeks	VAS A	8.0 {4.3–9.5}	5.5 {1.5–8.0}		

Design: study design type: NB = not blinded, SB = single-blinded (participants), DB = double-blinded (personnel), Cross = Cross-over. Population: ♀ = number of woman. Intervention: rTMS = repetitive transcranial magnetic stimulation. Location: anatomic region where intervention is applied. Outcomes of interest: THI = Tinnitus Handicap Inventory; VAS = Visual analogue scale 10 cm, NRS = Numeric rating scale, L = Loudness, A = Annoyance. P = pitch, At = Attention, MML = Minimum Masking Level, L(dB) = tinnitus loudness matching in decibels, Goebel Q = Goebel Questionnaire. “-“ = no data, () = Standard deviation, {} Interquartile range, Between-group analysis: (*) *p* < 0.05, (**) *p* < 0.01.

**Table 3 brainsci-13-00226-t003:** Overview of studies analysing the effect of acupuncture (AP) on tinnitus.

Author	Design	Intervention	N (♀)	Age(sd)	Controls	N (♀)	Age(sd)	Location	Intensity	Outcomes of Interest				
											Intervention		Control	
											T0	T1	T0	T1
Marks 1984 [55]	DB cross	eAP	14 (7)	51 (13)	Sham	14 (7)	51 (13)	Around ear distal points	20 min, daily 1 week	VAS L L(dB)	- 25 (-)	- 21 (-)	- 25 (-)	- 21 (-)
Axelssons 1994 [56]	SB cross	eAP	20 (0)	59 (-)	Sham	20 (0)	59 (-)	Around ear, distal points	3/week 5 weeks	VAS L VAS A VAS Aw	6.6 (-) 6.6 (-) 6.1 (-)	6.0 (-) 6.0 (-) 5.8 (-)	5.4 (-) 5.6 (-) 5.6 (-)	5.2 (-) 5.3 (-) 5.2 (-)
Vilholm 1998 [57]	DB	AP	29 (9)	52 (-)	Sham	25 (10)	54 (-)	Around ear and crown	30 min, 25 sessions	VAS L VAS A VAS Aw	6.5 (3.1) 7.9 (2.6) 6.9 (3.0)	6.2 (2.9) 7.4 (2.6) 6.5 (2.9)	6.7 (3.0) 7.7 (2.3) 6.5 (3.1)	6.7 (3.2) 7.8 (2.4) 6.7 (3.2)
Okada 2006 [58]	DB	AP	38 (-)	57 (12)	Sham	38 (-)	57 (12)	Temporal region	15 s, 1 session	VAS T **	Only *p*-value data were reported			
										Duration of relief		107 h		72 h
Wang 2010 [59]	NB	eAP	16 (4)	51 (4)	Sham	15 (0)	57 (2)	Head, distal points	1/week, 6 weeks	NRS6 T NRS4 L	Only *p*-value data were reported			
		AP	19 (0)	52 (3)										
Rogha 2011 [60]	DB	AP	27 (13)	46 (14)	Sham	27 (14)	49 (14)	Around ear, distal points	10 sessions	TSI ** VAS L *	46.9 (7.9) 8.9 (1.3)	31.7 (11.1) 5.3 (3.0)	46.6 (7.6) 8.7 (1.1)	42.9 (10.4) 7.5 (2.2)
Jeon 2012 [61]	DB	AP + infrared, education	17 (4)	47 (10)	Sham + infrared, education	16 (8)	49 (9)	Around ear and cervical	2/week, 5 weeks	THI VAS T	45.7 (25.6) 6.8 (1.8)	- -	39.8 (22.3) 6.7 (1.9)	- -
Laureano 2016 [62]	SB	AP	30 (15)	46 (10)	Sham	27 (23)	44 (13)	Around ear, distal points	20 min 12 sessions	THI * VAS A	48.0 (19.5) 6.2(2.4)	33.2 (17.5) 5.7 (2.5)	54.0 (18.4) 7.7 (2.1)	49.0 (22.2) 6.7 (2.4)
Doi 2016 [63]	NB	eAP	22 (14)	62 (-)	Waiting list	23 (13)	60 (-)	Temporal region	40 min, 2/week, 5 weeks	THI VAS L	56 {44–65.5} 8 {7–9}	Median 28 {8–55.5} 4 {3–6}	58 {48–76} 8 {7.5–9.5}	68 {46–76} 8 {8–10}
Kim 2017 [64]	NB	eAP (distal)	14 (-)	-	AP	13 (-)	-	Around ear, distal points	20 min, 2/week, 4 weeks	THI VAS L VAS A *	55.1 (24.4) 5.5 (2.4) 5.9 (2.6)	50.0 (27.4) 5.0 (2.3) 2.5 (2.4)	51.3 (24.8) 4.9 (2.2) 4.2 (3.0)	40.0 (33.7) 4.3 (2.5) 3.7 (3.0)
		eAP (Around ear)	15 (-)	-						THI VAS L VAS A *	35.4 (16.0) 4.8 (2.0) 4.3 (2.6)	33.9 (18.1) 4.5 (2.2) 4.3 (2.3)		
Naderinabi 2018 [65]	DB	AP	44 (18)	49 (1)	Sham	44 (17)	55 (8)	Around ear, distal points	3/week, 5 weeks	TSI ** VAS L **	43.8 (2.8) 9.6 (0.4)	23.1 (1.0) 2.3 (0.27)	43.5 (2.9) 9.5 (0.5)	33.1 (1.3) 7.8 (0.2)
Tu 2019 [34]	DB	AP	15 (8)	55 (12)	Sham	15 (10)	51 (10)	Head	6 sessions 3 weeks	THI	54.0 (-)	37.3 (-)	55.9 (-)	51.7 (-)
Kuzucu 2020 [23]	DB	AP	53 (34)	51 (10)	Sham	52 (35)	48 (11)	Around ear	2/week, 5 weeks	THI ** VAS A **	61.1 (12.7) 7.3(1.0)	40.3 (16.6) 3.7 (1.4)	59.3 (13.1) 7.0 (1.1)	60.7 (13.9) 6.9 (1.2)
Kim 2020 [66]	NB	AP	15 (3)	49 (8)	Tens	15 (2)	49 (23)	Around ear	2/week, 5 weeks	THI VAS L VAS A	41.7 (23.8) 5.7 (2.0) 5.3(2.1)	33.1 (10.6) 5.0 (1.7) 3.9 (1.6)	49.5 (22.9) 6.7 (1.7) 5.8 (2.5)	44 (12.3) 5.7 (1.6) 5.1 (1.3)
		eAP	15 (7)	46 (11)						THI VAS L VAS A	44.0 (19.0) 6.1 (1.8) 5.4 (1.8)	32 (14.5) 4.8 (1.3) 4.4 (1.6)		
Manz 2021 [26]	NB	Manual AP + usual care	26 (12)	53 (15)	usual care	24 (11)	46 (15)	Around ear, distal points	30 min 4 sessions 6 weeks	VAS L TFI	5.4 (2.4) 40.5 (24.4)	4.9 (2.6) 33.0 (25.7)	5.8 (2.3) 44.8 (23.1)	4.2 (2.5) 33.4 (21.5)

Design: study design type: NB = not blinded, SB = single-blinded (participants), DB = double-blinded (personnel), Cross = Cross-over. Population: ♀ = number of woman. Intervention: AP = acupuncture, eAP = electro acupuncture. Location: anatomic region where intervention is applied. Outcomes of interest: THI = Tinnitus Handicap Inventory; VAS = Visual analogue scale 10 cm, NRS = Numeric rating scale, L = Loudness, A = Annoyance, Aw = Awareness, T= tinnitus, L (dB) = tinnitus loudness matching in decibels, TSI = Tinnitus severity index, TFI = Tinnitus Functional Index, - = no data available, () = Standard deviation, {} = interquartile range, Between-group analysis: (*) *p* < 0.05, (**) *p* < 0.01.

**Table 4 brainsci-13-00226-t004:** Overview of studies analysing the effect of other physical interventions on tinnitus.

Author	Design	Intervention	N (♀)	Age	Controls	N (♀)	Age	Location	Intensity	Outcomes of Interest				
											Interval		Control	
											T0	T1	T0	T1
Rendell 1987 [67]	DB cross	Ultrasound	40 (16)	58 (7)	Sham	40 (16)	58 (7)	Proc. Mastoid	20 min, 1 session	NRS L		0.3 (-)		−0.6 (-)
										L(dB)		−0.4 (4.6)		−0.7 (6.4)
Erlandsson 1991 [68]	NB cross	TMD treatment	32 (-)	50 (10)	Bio-feedback EMG	32 (-)	50 (10)	General	15–30 min, 6 sessions versus 60 min, 10 sessions	VAS L NRS1-9 A	6.0 (-) 4.6 (-)	5.4 (-) 4.8 (-)	5.6 (-) 4.0 (-)	5.3 (-) 4.1 (-)
Podoshin 1995 [69]	NB	Biofeedback EMG	62 (28)	43 (-)	Ami-triptyline	40 (18)	46 (-)	General	30 min, 1/week, 10 weeks	NRS4 A *	-	−44%	-	−28%
Weise 2008 [37]	NB waiting list	Biofeedback + CBT	52 (23)	49 (12)	Waiting list	59 (26)	53 (12)	General	1 h, 12 sessions	TQ VAS A VAS L	54.8 (10.4) 5.0 (1.7) 5.7 (1.3)	32.5 (16.0) 4.2 (1.7) 4.4 (1.8)	55.0 (10.2) 4.9 (1.9) 6.1 (1.7)	49.5 (13.8) 5.2 (1.8) 5.7 (1.7)
Rocha et al. 2012 [70]	DB	MTP treatment	33 (-)	-	Sham	24 (-)	-	Neck, head, upper back	1/week, 10 weeks	THI * NRS10 L/A Nr sounds	Only *p*-value data were reported			
Atan 2020 [71]	DB	Kinesio tape	15 (8)	45 (11)	Sham	15 (9)	50 (11)	Cervical, ear, upper back	1/week, 4 weeks	THI VAS L	65.7 (10.8) 7.5 (1.2)	52.7 (13.0) 5.1 (1.3)	61.1 (17.7) 7.2 (2.3)	59.3 (17.8) 6.6 (1.9)
Delgado de la Serna 2020 [72]	SB	Manual + TMD treatment	30 (17)	44 (11)	TMD treatment	31 (19)	43 (12)	Cervical, ear, upper back	6 sessions 1 month	THI * VAS A *	36.1 (9.6) 6.8 (1.2)	14.4 (8.8) 2.8 (2.1)	34.2 (11.9) 6.7 (7.4)	28.3 (7.4) 4.7 (1.6)
Wal van der 2020 [24]	SB waiting	TMD treatment	40 (22)	46 (13)	Waiting list	40 (16)	45 (15)	Cervical, jaw	18 sessions, 9 weeks	TFI * TQ VAS L	55 (17) 37 (16) 4.8 (2.5)	41 (-) 32 (-) 4.2 (2.5)	48 (15) 34 (15) 4.7 (2.0)	43 (-) 34 (-) 4.8 (2.6)

Design: study design type: NB = not blinded, SB = single-blinded (participants), DB = double-blinded (personnel), Cross = Cross-over. Population: ♀ = number of woman. Intervention: MTP = myofascial trigger points, CBT = Cognitive behaviour therapy, TMD = temporomandibular disorder. Location: anatomic region where intervention is applied. Outcomes of interest: THI = Tinnitus Handicap Inventory; VAS = Visual analogue scale 10 cm, NRS = Numeric rating scale, L = Loudness, A = 10 Annoyance, L (dB) = tinnitus loudness matching in decibels, TQ = Tinnitus questionnaire, TFI = Tinnitus Functional Index, - = no data available, () = Standard deviation, Between-group analysis: (*) *p* < 0.05.

**Table 5 brainsci-13-00226-t005:** Risk of bias summary. Review authors’ judgments about each risk of bias item for each included study.

	Sequence Generation	Allocation Concealment	Blinding	Incomplete Outcome Data	Selective Outcome Reporting	Other sources of Bias	
Dobie, 1986 [42]							TENS
Kapkin, 2008 [43]						
Bonaconsa, 2010 [13]						
Lee, 2014 [44]						
Li, 2019 [30]						
Tutar, 2020 [45]						
Mirz, 1999 [46]							Laser
Nakashima, 2002 [47]						
Gungor, 2007 [48]						
Cuda, 2008 [49]						
Teggi, 2009 [50]						
Dejakum, 2013 [25]						
Mollasadeghi, 2013 [51]						
Ngao, 2014 [52]						
Thabit, 2015 [53]						
Demirkol, 2017 [54]						
Marks, 1984 [55]							AP
Axelssonsson, 1994 [56]						
Vilholm, 1998 [57]						
Okada, 2006 [58]						
Wang, 2010 [59]						
Rogha, 2011 [60]						
Jeon, 2012 [61]						
Doi, 2016 [63]						
Laureano, 2016 [62]						
Kim, 2017 [64]						
Naderinabi, 2018 [65]						
Tu, 2019 [34]						
Kuzucu, 2020 [23]						
Kim, 2020 [66]						
Manz, 2021 [26]							
Rendell, 1987 [67]							Ultrasound
Erlandsson, 1991 [68]							TMD treatment
Podoshin, 1995 [69]							Biofeedback
Weise, 2008 [37]							Biofeedback
Rocha, 2012 [70]							MTP
Atan, 2020 [71]							Kinesio tape
Wal van der, 2020 [24]							TMD treatment
Delgado de la Serna, 2020 [72]							TMD treatment
Low risk of bias							
Unclear							
High risk of bias							

## Data Availability

The database and data of analyses are available at osf.io. DOI: 10.17605/OSF.IO/KYB69.

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
