# Peer review of "Effects of Physical Interventions on Subjective Tinnitus, a Systematic Review and Meta-Analysis"

_brainsci, 2023, doi:10.3390/brainsci13020226_

Round 1

Reviewer 1 Report

The current systematic review and meta-analysis investigate the effect of interventions on tinnitus. The major concern is the reporting. The study should be reported according to PRISMA statement. Please find the following comments:

L 41-43 need references

Please define physical interventions 

The methods are insufficient and not informative. The study should be reported according to PRISMA statement. 

Please justify the rational for selection of the Hedge’s effect size rather weight mean difference or standardized mean difference.

The majority of figures have not the pooled effect size.

Author Response

Dear reviewer,

Thank you very much for your comments and suggestions. Due to the Christmas break, my co- authors were unavailable, which unfortunately delayed our response.

Please see the attachment for our reply and the changes we have made.

Yours sincerely,

Eric Bousema

Reviewer 2 Report

Bousema et al. performed a meta-analysis to assess the effectiveness of several physical intervention strategies for tinnitus. The authors meticulously identified and analyzed previous studies with well-described criteria and summaries. The meta-analysis will be a great contribution to clinicians and scientists.

My only comment is to ask the authors to consider indicating “bias risk” next to the Hedges’s g result, perhaps with symbols to help readers more easily determine potential bias. I do not mean to remove the already-existing bias risk table. Just that some information on that could be added to figure 4.

Author Response

(The authors gave the same response as above.)

Reviewer 3 Report

This systematic review has been prepared in accordance with all the necessary rules of systematic review and meta-analysis. Congratulations to the authors. The topic is very interesting, and the problem of tinnitus is widespread and increasingly common among the younger population, which is partly explained by the increasing acoustic load in the modern environment and numerous etiological factors arising from such a lifestyle. In the introductory part, the authors should highlight the problem of tinnitus in all age groups and the problem of etiological multifactoriality.

Author Response

(The authors gave the same response as above.)

Reviewer 4 Report

The aim of this systematic review and meta-analysis is to investigate the effects of physical interventions in patients with subjective tinnitus on measures of tinnitus perception. This review included 39 RCTs. The topic is interesting; however, this paper has different limitations and concerns listed below:

First, it’s not clear why the authors chose physical interventions, the included studies used different interventions. I think physical interventions are related to exercise or physical activity. For more clarity I suggest changing physical interventions in the title and the text, they could use non-pharmacological interventions and traditional interventions or physical therapy.

The second concern is that the authors ignore the PRISMA guidelines. Following PRISMA guidelines is highly recommended. For example, the methods section is general and the authors did not report it well.

The third concern regarding this review is the discussion. The discussion looks general, the authors did not focus on the main aim of their paper. They ignore discussing the main results and provide appropriate explanations and comparisons with previously published studies. In addition, the authors ignore the study limitations, any study has different limitations and these limitations should be addressed clearly. 

Author Response

(The authors gave the same response as above.)

Round 2

Reviewer 1 Report

-

Author Response

Dear reviewer,

Thank you very much for reviewing our manuscript. 

Yours sincerely, 

Eric Bousema

Reviewer 4 Report

I have no more comments, I recommend accepting this manuscript. 

Author Response

Dear reviewer,

Thank you for reviewing our manuscript and your positive recommendation.

Yours sincerely, 

Eric Bousema